# Haem transporter HRG-1 is essential in the barber's pole worm and an intervention target candidate

Yi Yang[1]☯, Jingru Zhou[1]☯, Fei Wu[1], Danni Tong[1], Xueqiu Chen[1], Shengjun Jiang[1], Yu Duan[1], Chaoqun Yao[2], Tao Wang[3], Aifang Du[1]*, Robin B. Gasser[3]*, Guangxu Ma[1,3]*

**1** Institute of Preventive Veterinary Medicine, Zhejiang Provincial Key Laboratory of Preventive Veterinary Medicine, College of Animal Sciences, Zhejiang University, Hangzhou, Zhejiang, China, **2** Department of Biomedical Sciences and One Health Center for Zoonoses and Tropical Veterinary Medicine, Ross University School of Veterinary Medicine, Basseterre, St. Kitts, West Indies, **3** Department of Veterinary Biosciences, Melbourne Veterinary School, The University of Melbourne, Parkville, Victoria, Australia

☯ These authors contributed equally to this work.
* afdu@zju.edu.cn (AD); robinbg@unimelb.edu.au (RBG); gxma1@zju.edu.cn (GM)

⊘ OPEN ACCESS

**Data Availability Statement:** All data necessary to replicate the findings is included in the manuscript without restriction.

## Abstract

Parasitic roundworms (nematodes) have lost genes involved in the *de novo* biosynthesis of haem, but have evolved the capacity to acquire and utilise exogenous haem from host animals. However, very little is known about the processes or mechanisms underlying haem acquisition and utilisation in parasites. Here, we reveal that HRG-1 is a conserved and unique haem transporter in a broad range of parasitic nematodes of socioeconomic importance, which enables haem uptake *via* intestinal cells, facilitates cellular haem utilisation through the endo-lysosomal system, and exhibits a conspicuous distribution at the basal laminae covering the alimentary tract, muscles and gonads. The broader tissue expression pattern of HRG-1 in *Haemonchus contortus* (barber's pole worm) compared with its orthologues in the free-living nematode *Caenorhabditis elegans* indicates critical involvement of this unique haem transporter in haem homeostasis in tissues and organs of the parasitic nematode. RNAi-mediated gene knockdown of *hrg-1* resulted in sick and lethal phenotypes of infective larvae of *H. contortus*, which could only be rescued by supplementation of exogenous haem in the early developmental stage. Notably, the RNAi-treated infective larvae could not establish infection or survive in the mammalian host, suggesting an indispensable role of this haem transporter in the survival of this parasite. This study provides new insights into the haem biology of a parasitic nematode, demonstrates that haem acquisition by HRG-1 is essential for *H. contortus* survival and infection, and suggests that HRG-1 could be an intervention target candidate in a range of parasitic nematodes.

## Author summary

Parasitic nematodes cannot synthesise haem *de novo* and must exploit haem from the environment or host animals. Mechanisms securing haem acquisition and its systemic distribution in these pathogens remain to be elucidated. In the present study, we identify a

**Funding:** This work is supported by National Natural Science Foundation of China (31602041 to YY, 32002304 to GM and 32172877 to AD), Zhejiang Province Public Welfare Technology Application Research Project, China (LGN20C180005 to YY), Natural Science Foundation of Zhejiang Province (LZ22C180003 to GM), National Key R&D Program of China (2017YFD0501200 to AD). RBG's research program is presently funded by the Australian Research Council, Yourgene Health and Phylumtech. The funders had no role in study design, data collection and analysis, decision to publish, or preparation of the manuscript.

**Competing interests:** The authors have declared that no competing interests exist.

unique haem transporter HRG-1 in a range of nematodes commonly found in animals, which shows haem binding activity *in vitro*, haem transporting activity in yeast, and an important role in haem homeostasis in the free-living model organism *Caenorhabditis elegans*. This transporter distributes in the basal laminae and interacts with a vacuolar ATPase in *Haemonchus contortus* (a paradigm and model to screen anthelmintic targets), enabling haem transport among tissues in this parasitic nematode. RNAi-mediated gene knockdown of *hrg-1* results in decreased viability of the infective larvae of barber's pole worm, which cannot establish infection or survive in the host animal. The uniqueness and essentiality of HRG-1 in this parasitic nematode provide novel insights into the haem uptake and distribution in helminths and a target candidate for the control of nematode infection.

## Introduction

Haem is an iron-containing porphyrin, which is essential for multiple cellular processes, including oxygen transfer, signal transduction and metabolism, in all life-forms [1,2]. Most eukaryotes synthesise haem *de novo* via a conserved, endogenous pathway involving eight enzymes [3,4], but recent evidence shows that roundworms (nematodes) have lost the genes that encode these enzymes during evolution and are, thus, entirely reliant on exogenous haem [5–7]. The loss of this *de novo* synthesis pathway raises questions about how nematodes acquire and transport haem from their surrounding environment, within or without their host. It is known that exogenous haem is imported into mammalian cells via membrane-bound haem transporters, including feline leukaemia virus subgroup c cellular receptor (FLVCR) and haem responsive gene-1 protein (HRG-1) [8–10]. Interestingly, the latter protein was first discovered in the free-living nematode, *Caenorhabditis elegans*–one of the best-studied metazoan model organisms, in which the roles of HRG-1 in haem acquisition, trafficking and homeostasis were elucidated [11], and three paralogues (HRG-4, HRG-5 and HRG-6) predicted. Apart from the identification of an HRG-1 orthologue in the parasitic nematode *Brugia malayi*–the causative agent of lymphatic filariasis of humans [12]–almost nothing is known about haem acquisition (i.e. transport into, within and between cells) in haematophagous and non-haematophagous parasitic nematodes [13–15]. Given that haem is essential for life, understanding the molecular and cellular biology of haem transporter(s) in these pathogens could lead to discovering ways of disrupting transporter function(s) for the purpose of blocking parasite development [5,7,13].

Here, we explore the extent of structural conservation of HRG-1 orthologues across a range of animal-parasitic nematodes, also with reference to their host animals; establish the distribution and functional roles of HRG-1 in a haematophagous nematode using complementary model organisms, including yeast and *C. elegans* as tools; and then evaluate this transporter protein as a (possible) novel anti-parasite intervention target candidate.

## Results

### Structural conservation of HRG-1 in nematodes, and distinctiveness from mammalian host orthologues

Using extensive, publicly available genomic and transcriptomic data sets [16,17], we identified nucleotide sequences coding for HRG-1 (haem transmembrane transporter) in 41 species of parasitic nematode (S1 Table). Manual curation of these sequences identified 1-to-1 *hrg-1* orthologues

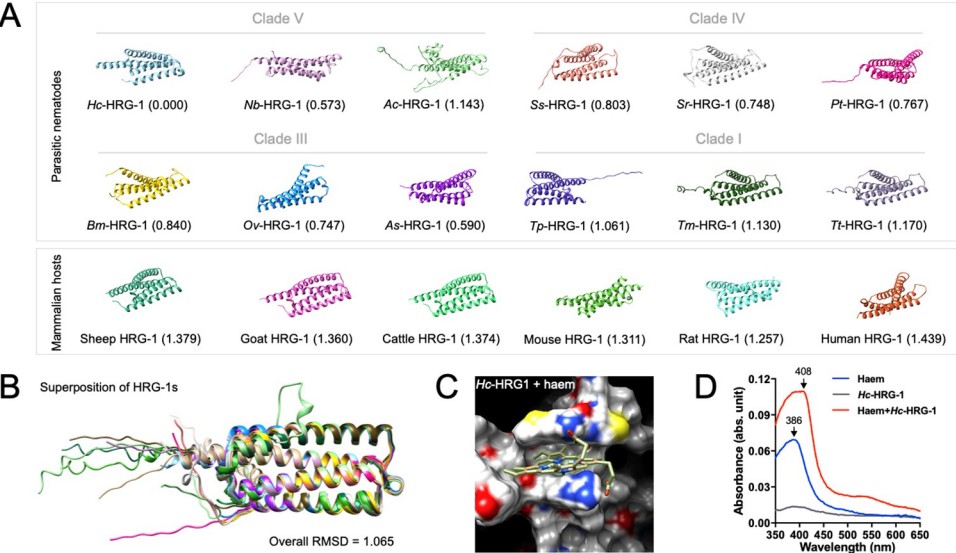

**Fig 1. Structural conservation of HRG-1 in parasitic nematodes and distinctiveness from mammalian host orthologues.** (A) Structures of HRG-1 predicted for parasitic nematodes and mammalian hosts using AlphaFold2. Structural alignment and comparisons are performed between *Hc*-HRG-1 of *Haemonchus contortus* and orthologues of *Nippostrongylus brasiliensis* (*Nb*-HRG-1), *Ancylostoma ceylanicum* (*Ac*-HRG-1), *Strongyloides stercoralis* (*Ss*-HRG-1), *Strongyloides ratti* (*St*-HRG-1), *Parastrongyloides trichosuri* (*Pt*-HRG-1), *Brugia malayi* (*Bm*-HRG-1), *Onchocerca volvulus* (*Ov*-HRG-1), *Ascaris suum* (As-HRG-1), *Trichinella pseudospiralis* (*Tp*-HRG-1), *Trichuris muris* (*Tm*-HRG-1), *Trichuris trichiura* (*Tt*-HRG-1), and mammalian host animals including sheep (UniProt ID: A0A6P3YH43), goat (A0A452EBG4), cattle (E1BKJ0), mouse (Q9D8M3), rat (A0A8L2R3J2) and human (Q6P1K1). Root mean square deviation (RMSD) values are shown to indicate the structural distinctiveness between *Hc*-HRG-1 of *H. contortus* and orthologues in other parasitic nematodes and animals. (B) Structural superposition of HRG-1s in *Caenorhabditis elegans* (Q21642) and orthologues in parasitic nematodes. An overall RMSD value is measured at 1.065 among the HRG-1 orthologues of the free-living *C. elegans* and selected parasitic nematodes *H. contortus*, *N. brasiliensis* and *A. ceylanicum* (clade V), *S. stercoralis*, *S. ratti* and *P. trichosuri* (clade IV), *B. malayi*, *O. volvulus* and *A. suum* (clade III), *T. pseudospiralis*, *T. muris* and *T. trichiura* (clade I). (C) *in silico* docking of haem and predicted three-dimensional structure of *Hc*-HRG-1 using AlphaFold2 [57]. (D) Maximum absorbance for haem (at a wavelength of 386 nm) and for haem + *Hc*-HRG-1 (at a wavelength of 408 nm).

in 39 nematodes of animals and 1-to-2 orthologues in two species, *Ancylostoma caninum* (hookworm) and *Dirofilaria immitis* (heartworm); thus, these parasitic nematodes had a reduced gene set compared with the free-living nematode *C. elegans* (n = 4). Comparative modelling revealed (relative) structural conservation of HRG-1 proteins encoded by single copy orthologs among these nematodes (clades I, III, IV and V; Root mean square deviation (RMSD) values ≤ 1.17), and a clear distinctiveness in both sequence and structure from orthologues in mammals (including human, mouse, rat and sheep; RMSD values ≥ 1.257) representing the host spectrum for these nematodes (Fig 1A), although the particular amino acid (aa) residues in sequences linked to haem transport were invariable among the nematodes and mammalian species studied (Fig 1B) [12,18]. Given the marked gene set reduction and structural conservation among nematodes, we elected to explore HRG-1 in the socioeconomically highly significant parasitic nematode, *Haemonchus contortus* (barber's pole worm), which is quite closely related to the model organism *C. elegans* (both being within clade V), allowing for comparative and complementary investigations.

## *Hc*-HRG-1 binds haem and enables transmembrane haem acquisition in an auxotrophic yeast model

After modelling the binding of *Hc*-HRG-1 to haem *in silico* (Fig 1C), we verified this binding *in vitro*. Compared with the absorbance peak at a wavelength of 386 nm for 10 μM haem in

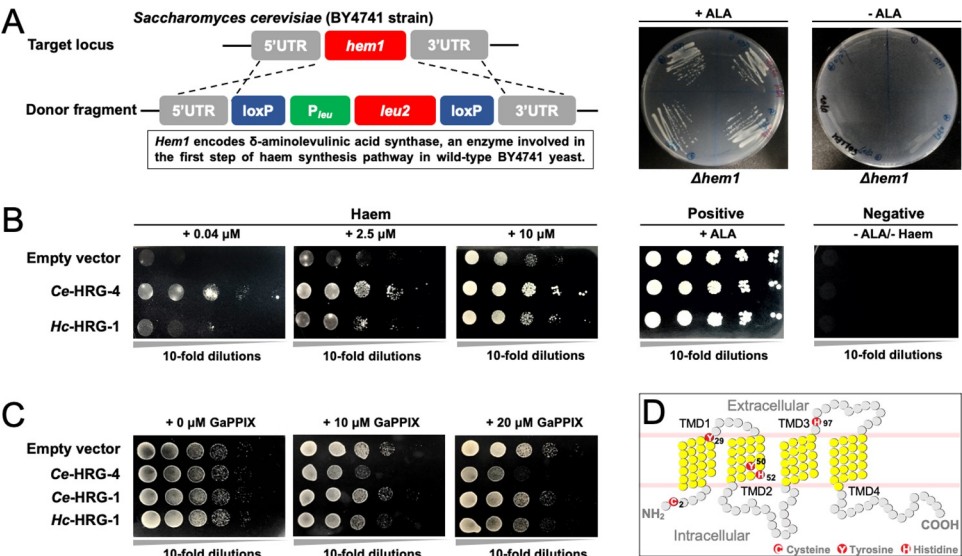

**Fig 2. *Hc*-HRG-1 mediates haem acquisition in a haem-deficient strain of yeast.** (A) The haem-deficient strain [*Δhem1*] of yeast is constructed by inserting δ-aminolevulinic acid synthase coding gene *leu2*, its promoter $P_{leu}$ and loxP sites (donor fragment) into the *hem1* locus of *Saccharomyces cerevisiae* BY4741 strain by homologous recombination. As *hem1* is involved in the first step of haem biosynthesis, *Δhem1* cannot synthesise endogenous haem *de novo* and cannot grow normally on plate unless supplemented with 5-aminolevulinic acid (ALA, an intermediate in the haem biosynthesis). (B) Haem spot growth assay of *Δhem1* transformed with empty parental vector (negative control), vector expressing HRG-1 paralogue in *Caenorhabditis elegans* (*Ce*-HRG-4) or HRG-1 orthologue in *Haemonchus contortus* (*Hc*-HRG-1). 10-fold serially diluted haem-depleted cells are spotted on plates supplemented with the indicated concentrations of haem. 250 μM ALA is used as positive control. Neither ALA nor haem is used in negative control. (C) Gallium protoporphyrin IX (GaPPIX, a toxic haem analogue) spot growth assay of yeast transformed with empty parental vector and vector expressing *Ce*-HRG-4 and *Ce*-HRG-1 of *C. elegans* or *Hc*-HRG-1 of *H. contortus*. Transformants are spotted by 10-fold serial dilutions on plates supplemented with 0, 10 or 20 μM GaPPIX. (D) Schematic diagram showing transmembrane domains (TMD indicated in yellow) and residues essential for haem transport (indicated in red) in the haem spot growth assay. Positions of essential residues are numbered.

phosphate-buffered saline (PBS), a mixture of haem and purified *Hc*-HRG-1 (10 μM) shifted the absorbance peak to wavelength 408 nm (Fig 1D). We further assessed the activity of this transporter in a haem-deficient strain of *Saccharomyces cerevisiae* (*Δhem1*; Fig 2A), which can neither synthesise endogenous haem without supplementation of 5-aminolevulinic acid (ALA, an intermediate in haem biosynthesis) nor acquire exogenous haem through cell membrane [19,20]. Yeast transformants expressing *Ce*-HRG-4 (the HRG-1 paralogue expressed in the apical plasma membrane) or *Hc*-HRG-1 grew better than the defect strain (empty vector control) on ALA-depleted medium when supplemented with haemin chloride, particularly at a low concentration (Fig 2B). HRG-1-mediated haem acquisition in *Δhem1* was further confirmed in a toxicity assay, in which wild-type yeast and transformants were exposed to gallium protoporphyrin IX (GaPPIX)–a toxic haem analogue (Fig 2C). Specifically, transformant expressing *Hc*-HRG-1 exhibited reduced growth in the presence of 10 μM and 20 μM GaPPIX, with reference to the control yeast containing empty vector (Fig 2C). Similar toxic effects were detected for both *Ce*-HRG-1 and *Ce*-HRG-4 transformants with reference to control yeast (Fig 2C), suggesting a relatively conserved haem acquisition-function of HRG-1 orthologues for clade V nematodes. Site-directed mutagenesis experiments confirmed functional conservation. By replacing individual residues (i.e. histidines, tyrosines and cysteines) in *Hc*-HRG-1 predicted to be crucial for haem transport with alanine [18,20,21], we showed that *Δhem1* transformants (carrying mutations C2A, Y29A, Y50A, H52A or H97A) exhibited decreased growth (S1 Fig), comparable to *Δhem1* transformed

with an empty vector, indicating that these critical residues direct haem binding and transport, consistent with findings for HRG-1 or HRG-4 in *C. elegans* [12,18].

### *Hc*-HRG-1 mediates the uptake of haem into intestinal cells of *C. elegans*

Here, we modelled haem acquisition in transgenic *C. elegans* using zinc mesoporphyrin (ZnMP)–a fluorescent haem analogue (Fig 3A). In the transgenic worms, we showed that heterologous *Hc*-HRG-1 fused with a green fluorescence protein (GFP) tag (P$_{Ce-hrg-1}$::*Hc*-HRG-1:: GFP) co-localised within intestinal cells to ZnMP, in a punctate manner (Figs 3B, subpanels a and b and S2), consistent with the cellular distribution of HRG-1 in *C. elegans* [18]. This result indicated transfer of haem to intracellular organelles, confirmed to be lysosome-related organelles based on co-localisation of ZnMP with lysosome-associated membrane protein (LMP) fused to GFP (P$_{Ce-lmp-2}$::*Ce*-LMP-2::GFP) in *C. elegans* (Figs 3C and S2). Subsequently, we explored the subcellular distribution of *Hc*-HRG-1, and showed in transfected HeLa cells that *Hc*-HRG-1::GFP was predominantly distributed in the plasma membrane (Dil-co-stained), in early and late endosomes, identified independently using RAB5A and RAB7A (Ras-related protein; specific marker for endosomes) fused to an mCherry tag, and lysosomes (Lyso-Tracker-stained) (Fig 4). These findings are consistent with the canonical movement of haem within a cell, which is usually mediated by HRG-4 (in the apical plasma membrane) and HRG-1 (in the subcellular compartment) in *C. elegans* [18,22], suggesting biological roles of the unique HRG-1 orthologue of parasitic nematodes in both haem uptake across plasma membrane and haem transport into subcellular compartments.

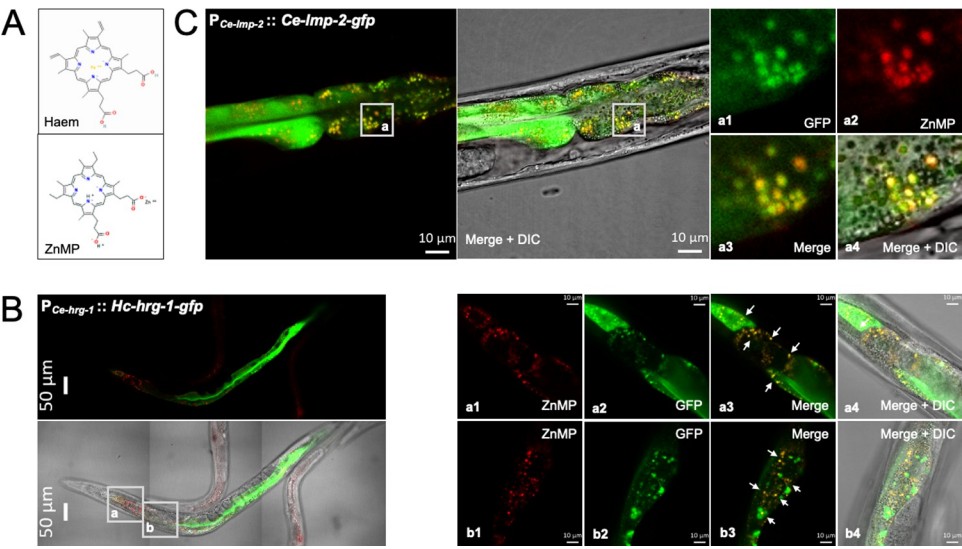

**Fig 3. Co-localisation of heterologous HRG-1 of *Haemonchus contortus* and a haem fluorescent analogue in the intestine of *Caenorhabditis elegans*.** (A) Structures of haem and the fluorescent zinc mesoporphyrinIX (ZnMP). (B) Heterologous expression of *Haemonchus contortus hrg-1* (*Hc-hrg-1*) in *Caenorhabditis elegans*. Expression of *Hc*-HRG-1 fused with a green fluorescent protein (GFP) tag in transfected *C. elegans* is driven by the endogenous promoter of *Ce-hrg-1* (P$_{Ce-hrg-1}$). Predominant distribution of *Hc*-HRG-1 is shown in the intestine of *C. elegans*, with punctate co-localisation of *Hc*-HRG-1 and ZnMP (indicated by white arrows). Details are shown in subpanels a (a1-a4) and b (b1-b4). (C) Co-localisation of ZnMP and a lysosome-associated membrane protein (*Ce*-LMP-2) fused with a GFP tag in the intestine of transfected *C. elegans*. Expression of *Ce*-LMP-2-GFP is driven by the endogenous promoter of *Ce-lmp-2* in this nematode. Details are shown in subpanels a (a1-a4) and b (b1-b4). DIC indicates images capture using differential interference contrast technique. Scale bars, 50 μm or 10 μm.

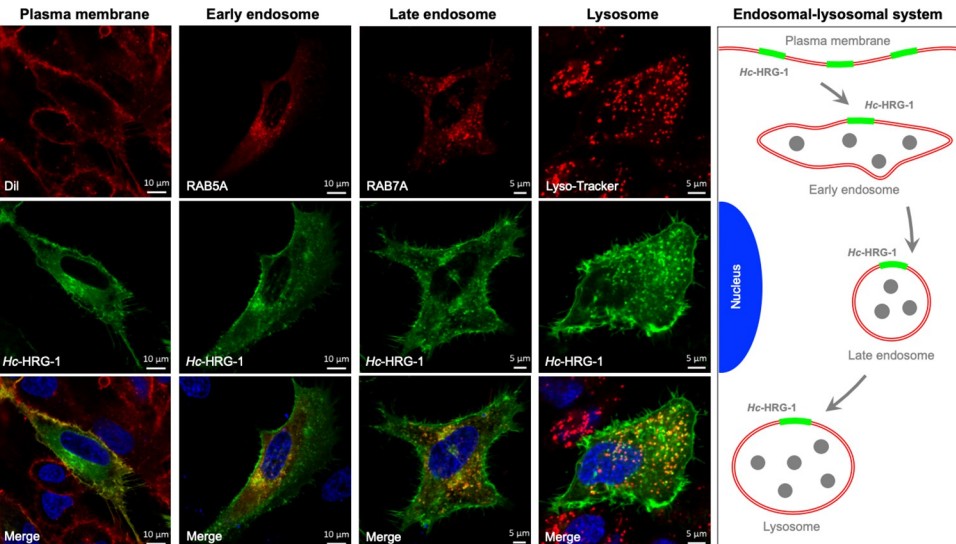

**Fig 4. Heterologous expression and localisation of *Hc*-HRG-1 in mammalian HeLa cells.** Some cells are transfected with *Hc-hrg-1-gfp*, then stained with Lyso-tracker Red (staining lysosomes) or DiI (staining plasma) and subjected to confocal microscopy analysis. Other cells are co-transfected with *Hc-hrg-1-gfp* and *hRab5a*-mCherry (a marker for early endosomes) or *hRab7a*-mCherry (a marker for late endosomes) plasmids and analysed using confocal microscopy. Blue staining indicates the nucleus while the yellow signal indicates the co-localisation of *Hc*-HRG-1-GFP and DiI, RAB5A, RAB7A or Lyso-Tracker in transfected cells. Scale bars (5 μm) are shown in the images. A schematic diagram summarising the cellular distribution of *Hc*-HRG-1 and its movement from plasma membrane to early endosome, late endosome and then lysosome is shown.

## *Hc*-HRG-1 interacts with an endo-lysosome-associated ATPase to enable haem trafficking in *H. contortus*

Here, we searched for molecules that interact with *Hc*-HRG-1 to facilitate haem uptake and transport in *H. contortus*. Yeast-two-hybrid (Y2H), glutathione-S-transferase (GST) pull-down and co-immunoprecipitation (co-IP) experiments (Fig 5A–5D) revealed a specific interaction between *Hc*-HRG-1 and a vacuolar H(+)-ATPase (V-ATPase) domain-containing protein (*Hc*-VHA-2) of *H. contortus*, with a punctate co-localisation of these two proteins detected in transfected HeLa cells (Fig 5E). The sequential deletion of each of the four transmembrane domains of *Hc*-HRG-1 showed that the third and fourth domains (representing the HRG superfamily; Fig 5F) are essential for this specific interaction with *Hc*-VHA-2 in HEK293T cells (Fig 5G), further indicating an involvement of HRG-1 and VHA-2 (representing lyso-some-related organelles) in haem trafficking in the barber's pole worm (see Fig 2), similar to that reported for mouse, rat and human cells [23].

## HRG-1 plays a role in haem homeostasis in both free-living and parasitic nematodes

We explored the transcription of the *hrg-1* gene in key developmental stages of *H. contortus*, and found high mRNA levels in infective (free-living, ensheathed L3) and parasitic (blood-feed-ing L4s and adult) stages (Fig 6A). After having predicted the haem-responsive element (HERE [24]) at 679 bp to 699 bp upstream of the transcriptional start site of *Hc-hrg-1* (S3A Fig), we assessed haem-responsive transcription of this gene by adding 20 μM or 100 μM of haemin chloride to cultures, and revealed a significant decrease in its transcription levels in *H. contortus* L1s/L2s ($P < 0.001$), adult females ($P < 0.001$) and adult males ($P < 0.001$), but no significant

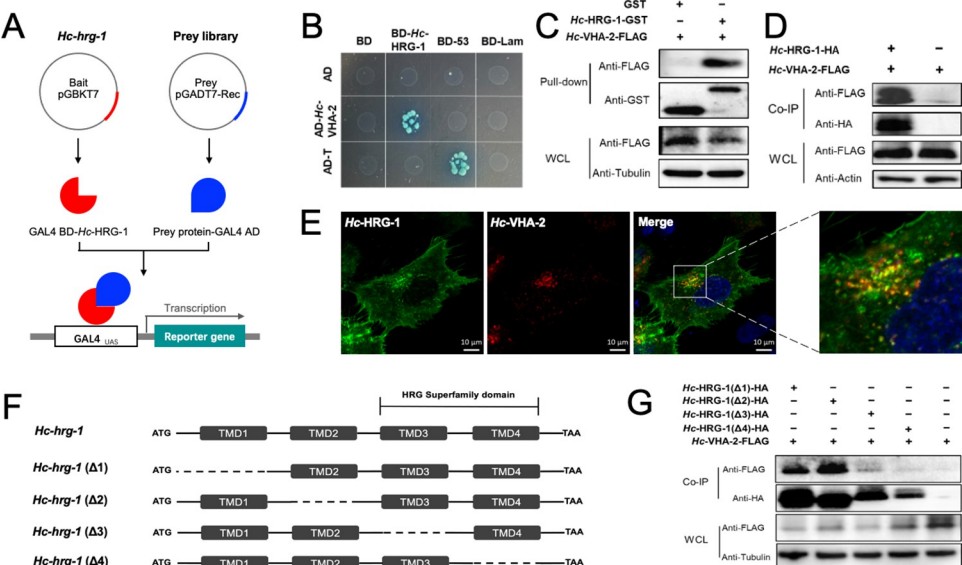

**Fig 5. Identification of a V-ATPase subunit interacting with HRG-1 in *Haemonchus contortus*.** (A) pGBKT7 vector expressing GAL4 transcription factor binding domain (BD)-*Hc*-HRG-1 and a pGADT7-Rec-cDNA library expressing prey protein-GAL4 activation domain (AD) are constructed for *H. contortus*. Complex of GAL4 BD-*Hc*-HRG-1 and prey protein-GAL4 AD initiates transcription of reporter gene. (B) Hybridisation between the prey (BD-*Hc*-HRG-1) and bait (AD-*Hc*-VHA-2) transformants. Hybridisation of Y187-pGADT7-T (AD-T) and Y2H-pGBKT7-lam (BD-lam) is used as negative control, whereas AD-T and Y2H-pGBKT7-p53 (BD-53) is used as blank control. Proliferating blue dots indicate interaction and non-proliferating white dots indicate no interaction. (C) Pull-down assay. The glutathione-S-transferase (GST) tag or *Hc*-HRG-1-GST fusion protein absorbed beads are incubated with the whole-cell lysate (WCL) of human embryo kidney cells (HEK293T) transfected with pcDNA3.1-*Hc*-VHA-2 fused with a FLAG tag, and analysed by Western blot using antibodies against GST and FLAG tags. (D) Co-immunoprecipitation (Co-IP) assay. HEK293T cells transfected with plasmids expressing *Hc*-HRG-1-HA (YPYDVPDYA-tag) and *Hc*-VHA-2-FLAG is subjected to immunoprecipitation (Co-IP) with anti-FLAG antibody-conjugated magnetic beads. The Co-IP and WCLs are individually analysed by Western blot using specific antibodies indicated. Actin is used as an internal control. (E) Co-localization of *Hc*-HRG-1-GFP and *Hc*-VHA-2-mCherry in HeLa cells. Cells are co-transfected with plasmids expressing *Hc*-HRG-1-GFP and *Hc*-VHA-2-mCherry fusion proteins and analysed using confocal microscopy after 24 hours. Blue staining indicates nucleus and yellow pixels indicates co-localization of the two proteins. Scale bar, 10 μm. (F) Domain mapping of *Hc*-HRG-1 that interacts with *Hc*-VHA-2. Schematic diagrams showing the domain architecture of *Hc*-HRG-1 and deletion (Δ1, Δ2, Δ3 and Δ4) of individual transmembrane domains (TMD). The TMD3 and TMD4 represent a HRG superfamily domain. (G) Co-IP assay indicates the TMD3 and TMD4 of *Hc*-HRG-1-HA are essential for its interaction with *Hc*-VHA-2-FLAG. Tubulin is used as an internal control. Scale bars, 50 μm or 10 μm.

in the L3 stage (Fig 6B); the latter finding was expected as this particular larval stage is ensheathed (S3B Fig), preventing the uptake of exogenous haemin chloride (either orally or through the cuticle) into the worm. The highest transcriptional level of *Hc-hrg-1* in the infective larvae of *H. contortus* represents a pre-adaptation to the infection/parasitism.

We verified the biological role of *Hc-hrg-1* in haem homeostasis using the haem acquisition model for *C. elegans*. In this model, *Hc-hrg-1* RNA interference (RNAi)-mediated (heterologous) gene knockdown resulted in a significant ($P < 0.01$) decrease in the *Ce-hrg-1* mRNA level, being comparable to homologous knockdown of *Ce-hrg-1* ($P < 0.01$) (Fig 6C). Compared with untreated *C. elegans*, the heterologous knockdown of *Ce-hrg-1* led to a decreased import of ZnMP into the intestine and subsequently a significant ($P < 0.001$) accumulation of ZnMP in the intestine, indicating interrupted haem utilisation at lysosome-related organelles in silenced *C. elegans* (Fig 6D and 6E). In addition, we also explored the functional relationship between *hrg-1* and *vha-2* in the haem homeostasis. In *H. contortus*, RNAi-mediated gene knockdown of *vha-2* ($P < 0.001$; Fig 6F) significantly increased the mRNA level of *hrg-1*

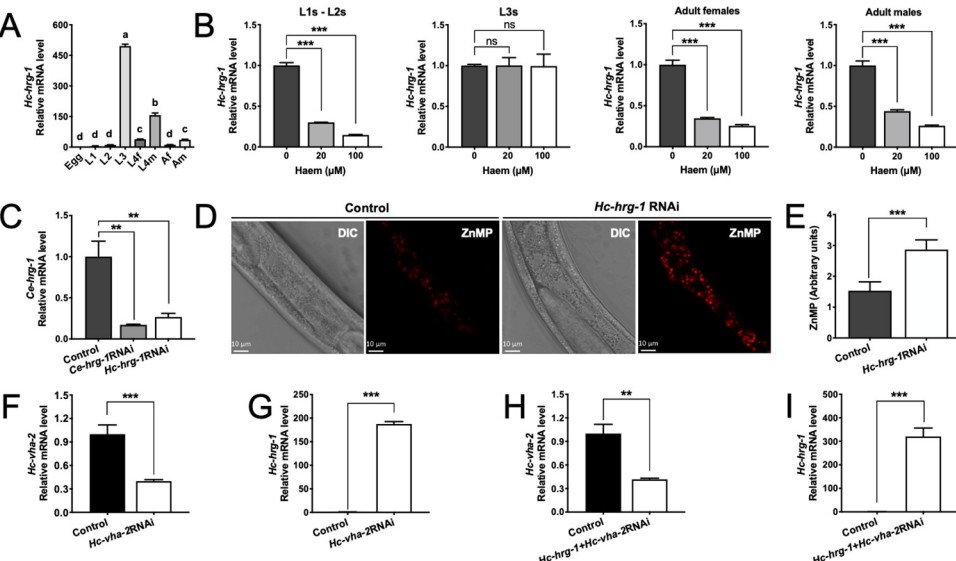

**Fig 6. Haem responsive transcription of *hrg-1* and its roles in haem homeostasis in *Haemonchus contortus*.** (A) Relative mRNA level of *hrg-1* in the egg, first- (L1s), second- (L2s), third- (L3s) and fourth-stage larvae (L4s), and adult female (Af) and adult male (Af) of *H. contortus*. *Hc-18sRNA* is used as an internal control. One-way ANOVA is used for statistical analyses. Different letter among data indicates a significant difference. (B) Relative mRNA level of *hrg-1* in L1s/L2s, L3s, adult females and adult males of *H. contortus* when exposing to 0 μM, 20 μM and 100 μM of hemin chloride. (C) Relative mRNA levels of *Ce-hrg-1* in *C. elegans* N2 worms fed with bacteria expressing double stranded RNA targeting *cry1Ac* of *Bacillus thuringiensis* (irrelevent control), *Ce-hrg-1* of *C. elegans* (*Ce-hrg-1*RNAi) and *Hc-hrg-1* of *H. contortus* (*Hc-hrg-1*RNAi). *Ce-actin-1* is used as an internal control. (D) *Hc-hrg-1*RNAi mediated gene knockdown of *Ce-hrg-1* results in accumulation of zinc mesoporphyrin IX (ZnMP; a fluorescent haem analogue) in the intestine of treated worms, compared with untreated worms (Control). DIC, differential interference contrast. Scale bar, 10 μm. (E) Quantification of ZnMP using ImageJ software. (F) Relative mRNA levels of *Hc-vha-2* in *H. contortus* fed with bacteria expressing double stranded RNA targeting *cry1Ac* of *Bacillus thuringiensis* (*Bt-cry1Ac*; irrelevent control) and *Hc-vha-2* (*Hc-vha-2*RNAi). (G) Relative mRNA levels of *Hc-hrg-1* in *H. contortus* fed with bacteria expressing double stranded RNA targeting *Bt-cry1Ac* (irrelevent control) and *Hc-vha-2*. (H) Relative mRNA levels of *Hc-vha-2* in *H. contortus* fed with bacteria expressing double stranded RNAs targeting *Hc-hrg-1* and *Hc-vha-2*. (I) Relative mRNA levels of *Hc-hrg-1* in *H. contortus* fed with bacteria expressing double stranded RNAs targeting *Hc-hrg-1* and *Hc-vha-2*. Data are showed as mean ± standard deviation (SD), $n = 10$. A $2^{-\Delta\Delta CT}$ method is used for relative transcriptional data normalisation. Data are showed as mean ± standard deviation, $n \geq 3$. Student t-test is performed for statistical analyses. ns: non-significant, $^*P < 0.05$, $^{**}P < 0.01$, $^{***}P < 0.001$.

compared with untreated controls ($P < 0.001$; Fig 6G). Simultaneous RNAi of both *hrg-1* and *vha-2* did not increase the knockdown effect on *vha-2* (Fig 6H), but led to a significant increase ($P < 0.001$) in the *hrg-1* mRNA level (Fig 6I).

## HRG-1 expression profile in tissues differs between free-living and parasitic nematodes

Heterologous expression of P$_{Ce-hrg-1}$::*Hc*-HRG-1::GFP was detected in the apical membrane of the anterior intestine, apical and basal membranes of medial intestine, and basal membrane and cellular compartments of posterior intestine of *C. elegans* (Figs 7A and S2). By contrast, using indirect immunofluorescence chemical analysis, *Hc*-HRG-1 was observed in the basal lamina of the gonads, hypodermis and, as expected, the intestine (with a weak signal detected in the epithelium), in adult *H. contortus* (irrespective of sex) (Fig 7B). The broader tissue expression pattern of HRG-1 in *H. contortus* than that in *C. elegans* indicates a distinctive role of this protein in the systemic haem homeostasis in the blood-feeding parasitic nematode, likely involving exporters for haem trafficking among tissues/organs [25–28]. The distribution

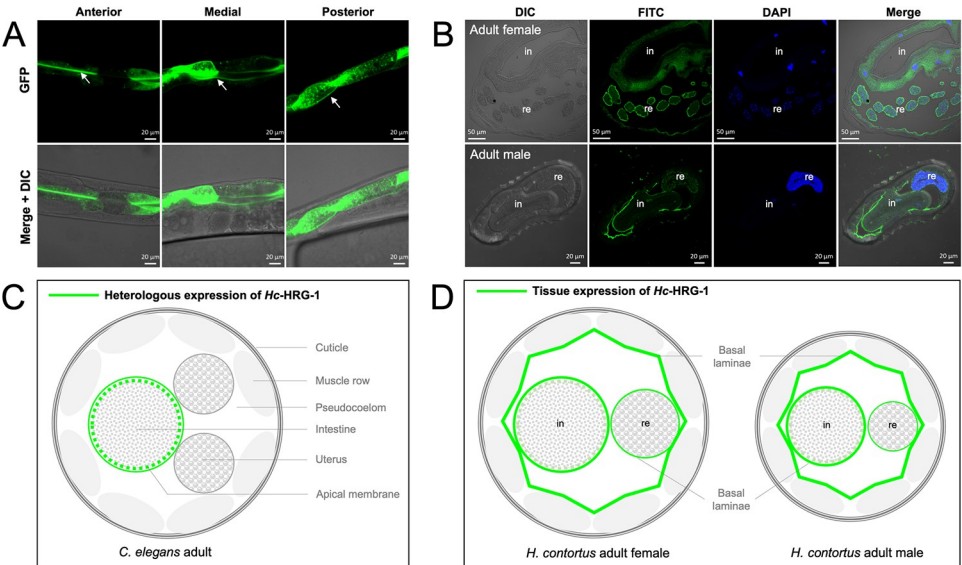

**Fig 7. Tissue expression of HRG-1 in *Caenorhabditis elegans* and *Haemonchus contortus*.** (A) Heterologous expression of *H. contortus* HRG-1 fused with green fluorescent protein (GFP) in the anterior, medial and posterior intestinal tract of *C. elegans*. (B) A schematic diagram indicating specific HRG-1 expression on the apical and basal membrane of intestine in *C. elegans*. No distribution of HRG-1 is found in other tissues. (C) Expression of HRG-1 at the basal laminae that covering both intestine (indicated by in), reproductive tract (indicated by re) and muscle of adult female and male *H. contortus*. 4',6-diamidino-2-phenylindole (DAPI) stains nucleus in blue. Scale bar, 50 μm or 20 μm. (D) A schematic diagram indicating a broad tissue expression (particularly at basal membranes) of HRG-1 in adult worms (female and male) of *H. contortus*.

of HRG-1 in the basal laminae of cells in various organs in *H. contortus* (Fig 7C and 7D) likely relates to a substantial uptake of large amounts of free haem originating from lysed erythrocytes from host blood–usually associated with major toxicity [14,29,30]. Interestingly, in adult *H. contortus*, while *Hc*-HRG-1 was not predominantly detected in the male reproductive tract, it was abundant in the uterus of the female (Fig 7B and 7D), suggesting that this haem transporter is integral to egg production and/or pre-embryonic development.

## HRG-1-mediated haem acquisition and homeostasis is essential for *H. contortus*

Using a feeding approach, RNAi resulted in a significant reduction of the mRNA levels of *Hc-hrg-1* in *H. contortus* on both days 3 (L2s; $P < 0.01$) and 7 (L3s; $P < 0.001$), compared with the untreated control (Fig 8A). Efficient knockdown of *Hc-hrg-1* significantly reduced the toxic effect of 5 μM and 20 μM of GaPPIX (a toxic haem analogue) on larvae ($P < 0.05$) after four days of treatment, compared with control (Fig 8B–8D), indicating that RNAi-mediated knockdown of *Hc-hrg-1* affected the uptake of GaPPIX via HRG-1 into *H. contortus*, supporting the proposal that this protein functions in haem acquisition. In addition, RNAi-mediated knockdown of *Hc-hrg-1* in *H. contortus* larvae resulted in sick (exhibiting retarded development, limited motility and viability) or lethal phenotypes. Specifically, increased numbers of sick (30%; $P < 0.001$) and dead larvae (24%; $P < 0.01$) were found after three days of RNAi treatment, compared with untreated control worms (~12%) (Fig 8E and 8F), and the percentages of sick (18%; $P < 0.001$) and dead larvae (38%; $P < 0.01$) increased significantly after four more days (Fig 8G and 8H). With reference to untreated controls, the supplementation of exogenous haem (5 μM and 10 μM) to the culture medium compromised somewhat the *Hc-hrg-1*

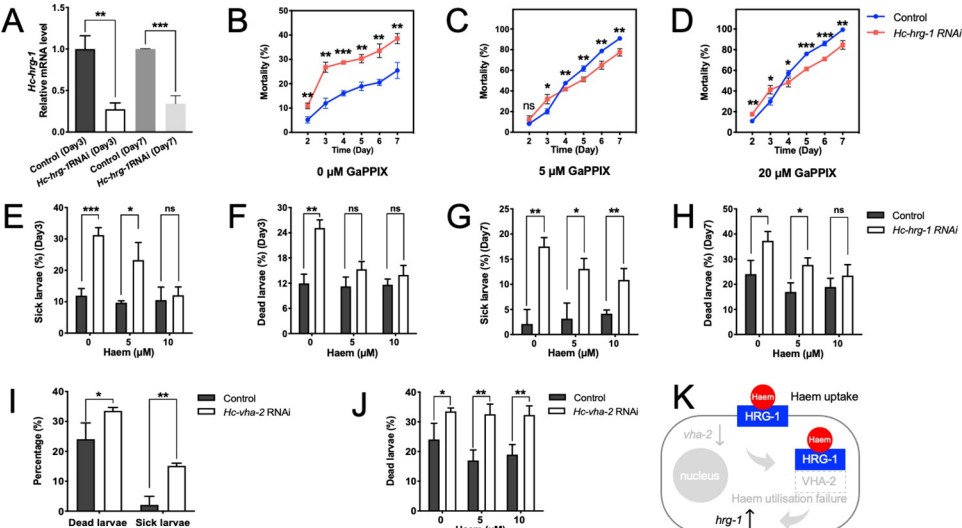

**Fig 8. RNA interference mediated gene knockdown of *hrg-1* results in compromised haem uptake in and survival of *Haemonchus contortus*.** (A) Relative mRNA levels of *Hc-hrg*-1 in *H. contortus* larvae fed with bacteria expressing double stranded RNA targeting *cry1Ac* of *Bacillus thuringiensis* (irrelavent control) and *Hc-hrg-1* after three (Day3) and seven days (Day7). *Hc-18sRNA* is used as an internal control. (B-D) RNAi-treated worms are fed with 0 μM, 5 μM and 10 μM of Ga (III) complex of the haem precursor protoporphyrin IX (GappIX; a toxic haem analogue) for 6 days, and subjected to mortality analysis every day. (E-H) Percentages of sick (low vitality, developmental retarded or deformed) and dead larvae are found in *H. contortus* fed with bacteria expressing double stranded RNA targeting *Hc-hrg-1* at day 3 and 7, compared with that targeting irrelevent control. Supplementation of 5 μM or 10 μM of haem into culture medium completely or partilly rescue the sick and lethal phenotype of treated larvae. (I) Percentages of dead and sick (low vitality, developmental retarded or deformed) larvae after *Hc-vha-2*RNAi measured at day 7. (J) Percentages of dead larvae after *Hc-vha-2*RNAi treatment with supplementation of 0 μM, 5 μM and 10 μM hemin chloride. Treatment with *Bt-cry1Ac*RNAi is used as irrelavant control. *Hc-18sRNA* is used as an internal control. A $2^{-\Delta\Delta CT}$ method is used for relative transcriptional data normalisation. Data are showed as mean ± SD, $n \geq 3$. Student t-test is performed for statistical analyses. ns, no significance, $^*P < 0.05$, $^{**}P < 0.01$, $^{***}P < 0.001$. (K) A schematic diagaram indicating the roles of *Hc-hrg-1* and *Hc-vha-2* in haem homeostasis and utilisation in *H. contortus*. Low transcriptional level of *Hc-hrg-1* do not affect the transcription of *Hc-vha-2*, whereas lower mRNA level upregulates the transcription of *Hc-hrg-1*.

RNAi-associated phenotype (sick and dead), achieving an enhanced effect at the higher concentration (Fig 8E–8H). These findings suggest the essentiality of *Hc-hrg-1* and the associated haem acquisition and homeostasis in the larvae development and survival of *H. contortus*.

We also explored the essentiality of *Hc-vha-2* in this blood-feeding parasitic nematode. Although knockdown of *vha-2* also resulted in increased numbers of sick (by ~ 12%; $P < 0.01$) and dead (by ~ 10%; $P < 0.05$) larvae, it was lower than for *hrg-1* RNAi-treated larvae (Fig 8I). This result is expected as knockdown of *vha-2* increased the level of *hrg-1* mRNA in *H. contortus* (Fig 6F–6I), further supporting the functional relationship between *hrg-1* and *vha-2* in haem homeostasis. However, in contrast to results for rescued *hrg-1* RNAi-treated larvae, the lethal phenotype caused by *vha-2* RNAi could not be rescued by adding exogenous haem (5 μM or 10 μM) to culture medium (Fig 8J), supporting a negative relationship between VHA-2-mediated haem utilisation and *hrg-1* transcription within cells and suggesting an involvement of VHA-2 in haem homeostasis downstream of HRG-1 in *H. contortus* (Fig 8K). These findings indicate that *hrg-1* is essential for life in *H. contortus*.

## Gene-silenced *H. contortus* larvae do not establish in sheep

We assessed larval infectivity by inoculating helminth-free sheep with: (a) *H. contortus* L3s raised from *hrg-1* RNAi-treated L1s/L2s; (b) with wildtype (normal) infective L3s; or (c) with

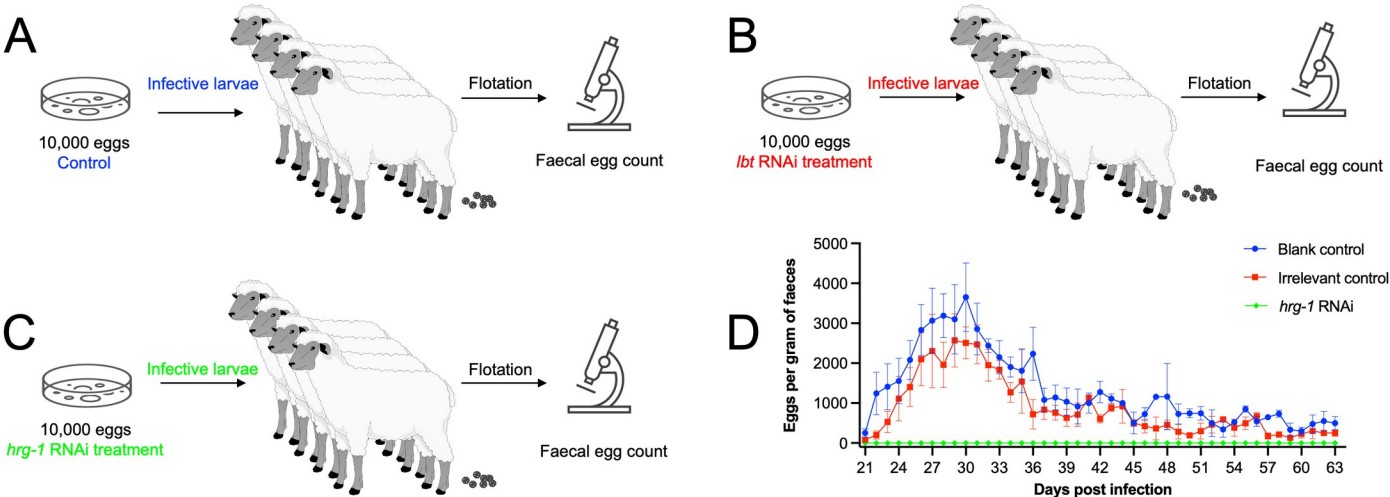

**Fig 9. Animal infection with RNA interference treated infective larvae of *Haemonchus contortus*.** (A-C) Eggs (n = 10,000) of *H. contortus* are collected from faeces, hatched on plates and fed with faecal matter (Blank control) or bacteria expressing double stranded RNA targeting *cry1Ac* of *Bacillus thuringiensis* (*Bt-cry1Ac*; irrelevant control) or *Hc-hrg-1* (*hrg-1*RNAi) for 7 days. Infective larvae of these treated worms are used to infect helminth-free sheep. Faeces are collected from the infected sheep every day from 21 days post infection to 63 days post infection to perform worm egg counting using a flotation method. (D) The numbers of eggs per gram (EPG) of faeces collected from sheep of blank control, irrelevant control and *hrg-1*RNAi groups are calculated and shown as mean ± standard error of the mean (*n* = 4) in the line graph. Images in this figure are drawn by the authors.

L3s derived from L1s/L2s treated by RNAi, targeting an irrelevant gene (*cry1Ac* of *Bacillus thuringiensis*) (Fig 9). While patent *H. contortus* infection established for (b) and (c), no eggs were detected in the faeces of sheep infected with *hrg-1* "knocked-down" larvae throughout the entire experimental period of 63 days (a) (Fig 9), and no adult worms were detected at *post mortem* on day 63 in the abomasa of sheep inoculated with "knocked-down" larvae.

## Discussion

Nematodes are a very diverse and large group of animals that inhabit almost every ecosystem, and can be classified into five distinct clades (I to V), many of which are parasites of mammals [31,32]. Some species, such as *Haemonchus*, *Ancylostoma* and *Necator*, are blood-feeding worms, with direct life cycles, and others (e.g., within the spiruroid and filarioid groups) are transmitted to their definitive animal hosts via arthropod vectors (e.g., black flies or mosquitoes) [32]. Not only are these parasites interesting from a biological perspective, in that they maintain a very intimate relationship with the host that they infect, and rely heavily on maintaining a balance in this relationship, so that neither parasite nor host is disadvantaged in a major way, and with the parasites finding ways of acquiring nutrients from the host to survive and/or modulating or suppressing the host immune response(s) to evade or avoid direct immune attack. Despite this intricate host-parasite relationship, many parasitic nematodes, such as blood-feeding nematodes, can cause devastating diseases in humans and animals [33,34]; they deprive the host of blood, causing anaemia and associated symptoms (e.g., impaired intellectual and physical development, and reduced economic productivity), and can lead to death, particularly in young individuals. Given the socioeconomic impact that these parasites cause in animal and human populations [35,36], and the challenges associated with their control, particularly relating to resistance to, the inefficacy of, current treatments and lack of vaccines [37,38], there is a need to find better ways of controlling parasitic nematode infections; this could be achieved through finding the Achilles heel in these nematodes to disrupt or interrupt biological processes or pathways. For these reasons, our research mission has

been focused on understanding the biology and biochemistry of socioeconomically important parasitic nematodes, with an emphasis on finding new intervention targets.

Here, we elucidated the structure and function of the haem transporter (HRG-1) in one of the most pathogenic, blood-feeding nematodes of the gastrointestinal tract of ruminants–*H. contortus* (barber's pole worm) using various complementary experimental tools (e.g., yeast, *C. elegans* and mammalian cells) and molecular assays (e.g., heterologous expression and RNAi). This transporter is of particular interest, because many parasite genera (including *Brugia*, *Haemonchus* and *Necator*) have lost genes that encode enzymes involved specifically in endogenous haem biosynthesis [6,7,12], such that they are completely reliant on acquiring exogenous haem from the environment within or outside of their host, depending on their developmental stage, which means that this transporter is critical for the survival of the parasite, thus representing an Achilles heel.

Indeed, compared with free-living nematode *C. elegans*, which has *hrg-1* and three related paralogues (i.e., *hrg-4*, *hrg-5* and *hrg-6*), the blood-feeding nematode *H. contortus* has only one *hrg-1* gene, representing an intervention target. Other parasitic nematodes studied–for which molecular data sets were available–also have only such *hrg-1* gene. Being responsible for the acquisition of exogenous haem and nematode survival, the gene product (HRG-1) is structurally conserved among nematode species studied but distinct from the orthologous host molecule(s), indicating its selectivity from the host animal. These features, together with evidence that third-stage larvae of *H. contortus* whose *hrg-1* gene had been "knocked down" did not establish or survive within the permissive host animal, indicating that HRG-1 is indeed a promising intervention target. Given that HRG-1 is a membrane-bound protein expressed in the nematode's intestine, it could represent a novel vaccine target, since haem-binding glutathione transferases have been exploited in vaccine development against hookworm infection in humans [39–41], particularly if parenteral immunisation with recombinant *Hc*-HRG-1 or a sub-component thereof can stimulate protective immunity against *H. contortus* [42]. This aspect deserves exploration, with a focus on achieving longer lasting and more consistent immunoprotection in all age-groups and species of host animals (sheep and goats) than achievable using the commercially available vaccine (Barbervax) produced from native gut-derived molecules from *H. contortus* [43–47].

Questions surrounding *hrg-1* and associated haem acquisition in parasitic nematodes that still warrants investigation in the future include: (i) Why are *hrg-1* paralogues absent from both blood-feeding and non-blood-feeding parasitic nematodes?–given that these genes are haem responsive and theoretically a complement for the lack of haem synthesis in free-living and parasitic nematodes [5–7,11,12]. (ii) How do nematodes regulate haem trafficking from haem-enriched intestinal lumen/cells?–given that the unique haem transporter is located predominantly in the outer layer of intestine of *H. contortus*–as distinct from that of *C. elegans* [18,22,48]. (iii) Why did the *hrg-1* gene-silenced larvae fail to establish infection, parasitism and/or reproduction in host animals?–given that ~ 40% of the RNAi-treated larvae appeared normal *in vitro*. (iv) Might *hrg-1* be evaluated as a RNAi-based interventiaon target or a immunogen for vaccine design?–given that amino acid residues linked to haem transport are invariable among the parasitic nematodes and mammalian species examined thus far. Addressing these questions will not only help understand the functional role of *hrg-1* and its gene product, but might assist in developing a new intervention approach against socioeconomically important parasitic nematodes.

## Materials and methods

### Ethics statement

All animal experimentation was approved by the Experimental Animals Ethics Committee of Zhejiang University (permit no. 20170177), Hangzhou, People's Republic of China.

### Nematodes and cell lines

Wild-type *C. elegans* (N2 strain) was obtained from the *Caenorhabditis* Genetics Center (CGC) and maintained according to the established protocols. *H. contortus* (ZJ strain) was routinely maintained in Hu sheep using established method [49]. Eggs, L1s, L2s, L3s, L4s and adults of *H. contortus* were produced and isolated using standardised techniques [49,50]. *Spodoptera frugiperda* insect cells (Sf9), HeLa and HEK293T cells and *S. cerevisiae* BY4741 strains (*MATa*, *his3Δ1*, *leu2Δ0*, *met15Δ0* and *ura3Δ0*; Scientific Research and Development GmbH, Oberursel, Germany) were stored and cultured following manufacturer's instructions.

### Species tree and tree based on *hrg-1* orthologues

Genomic and transcriptomic data sets available for major parasitic nematodes at WormBase ParaSite (https://parasite.wormbase.org/species.html; version: WBPS15) were used for the genome-wide identification of genes coding for haem transporter domain-containing protein. In brief, the domain architecture of deduced amino acid sequences of parasitic nematodes was analysed to predict proteins of the haem transporter HRG family using InterProScan v.5.24.63 [51]. This prediction was confirmed by searching the candidate amino acid sequences against NCBI CDD database [52]. Putative orthologues of *hrg-1* of *C. elegans* in parasitic nematodes and mammalian hosts were predicted using reciprocal BLAST searching and employing Ensembl Compara database [53]. Manually curated amino acid sequences of HRG-1 orthologues were aligned using MUSCLE, and aligned sequences were used to phylogenetic analysis employing MEGA X v.10.2.5 [54–56].

### Structure prediction and docking

The structures of HRG-1 orthologues were predicted using AlphaFold v. 2.1.0, or retrieved from AlphaFold Protein Structure Database [57,58]. Molecular docking of HRG-1 orthologues with haem was performed using the ClusPro server [59,60]. Binding sites, active sites and structural alignment of HRG-1 orthologues were studied and displayed using the PyMOL molecular graphics system v2.5 [61].

### Haem binding assay

The complementary DNA sequence of *hrg-1* from *H. contortus* (*Hc-hrg-1*) was cloned and sequenced using a First Strand cDNA Synthesis Kit (Toyobo Co., Ltd., Osaka, Japan) according to the manufacturer's instructions. The coding sequence of *Hc-hrg-1* with a His tag at the C-terminus was inserted into the pFastBacHT B vector in the *EcoR* I site, which was subsequently tranformed into *Escherichia coli* DH10Bac (Beyotime Biotechnology) to produce recombinant bacmids. Recombinant bacmids (confirmed by direct sequencing) were transfected into sf9 cells using LipoInsec Transfection Reagent, according to the manufacturer's instructions (Beyotime Biotechnology). Oligonucleotide primers used for PCR-based molecular cloning are shown in S2 Table. Recombinant *Hc*-HRG-1 protein was purifed using a Ni-NTA agarose column (Qiagen, Beijing, China), and protein amount determined using the Bradford Protein Quantitation Kit (Fude Biological technology, Hangzhou, China). The haem binding assay was performed as described previously [15]. In brief, haemin chloride (10 μM) was mixed with *Hc*-HRG-1 (10 μM) and incubated at room temperature, shaking for 2 h. Haem binding was measured from three technical replicates by absorption in a Synergy HT Multimode Reader (BioTek, USA), and the mean value calculated.

## Heterologous expression

Using the NovoRec PCR Seamless Cloning Kit (Novoprotein, Shanghai, China), the promoter of *Ce-hrg-1* (WBGene00019830) (2 kb upstream of the start codon, ATG) and the coding sequence of *Hc-hrg-1* were cloned into the *Sal* I and *Kpn* I sites of the vector pPD95.75, respectively, to create a P*ce-hrg-1*::*Hc*-HRG-1::GFP expression vector. The transgenic worms were made by microinjection of 50 ng/μl of the P*ce-hrg-1*::*Hc*-HRG-1::GFP plasmid, and positive selection marker PRF4 plasmid was microinjected into the wild-type N$_2$ strain of *C. elegans*. Expression of *Hc*-HRG-1::GFP in transgenic worms was examined using a confocal microscope (LSM780, Carl Zeiss). Oligonucleotide primers used for PCR-based molecular cloning and expression are shown in S2 Table.

## ZnMP uptake assay

The fluorescent haem analogue Zn(II) Mesoporphyrin IX (ZnMP) uptake assay was performed using an established protocol with minor modification [11]. In brief, larvae (F1) of transgenic *C. elegans* were exposed to 20 μM ZnMP (Frontier Scientific, Logan, USA) for 16 h in mCeHR-2 medium containing 1.5 μM of haem. Fluorescence intensity of ZnMP was measured by confocal microscopy at a fixed excitation of 488 nm and emission of 575 nm. Co-localisation of ZnMP and P*ce-hrg-1*::*Hc*-HRG-1::GFP in transgenic worms was conducted; the fluorescence intensity in worms was quantified using ImageJ (v.1.50 i, Wayne Rasband, National institutes of Health, USA).

## Subcellular localisation

The coding sequence of *Hc-hrg-1* was cloned into the pLentiCMV-EGFP-Puro vector at the *BamH* I site, and *hRab5a* (gene bank number: NM004162) and *hRab7a* (gene bank number: NM004637.5) were individually cloned into the pLentiCMV-mCherry-Puro vector in the *BamH* I site. Plasmid (2 μg) was transiently transfected into HEK293T or HeLa cells in 6-well plates or laser confocal Petri dishes using lipofectamine 2000 (Invitrogen, USA) using the manufacturer's protocols. Oligonucleotide primers used for PCR-based molecular cloning and eukaryotic expression are shown in S2 Table. The transfected cells were incubated in Lyso-Tracker Red (Beyotime Biotechnology, Shanghai, China) diluted to 1:20,000 in Dulbecco's modified Eagle medium (DMEM) supplemented with 5% foetal bovine serum (FBS) at 37°C in 5% CO$_2$ for 20–30 min, to label lysosomes. Cell nuclei were stained using 4',6-diamidino-2-phenylindole (DAPI; Beyotime Biotechnology) and plasma membrane was stained using 1,1'-dioctadecyl-3,3,3',3'-tetramethylindocarbocyanine perchlorate (DiI) staining kit (Beyotime Biotechnology) according to the manufacturer's protocol. Subcellular localisation was determined by confocal microscope (LSM780, Carl Zeiss).

## Yeast spotting assay and site-directed mutagenesis

A haem-deficient strain [*Δhem1*] of yeast (*S. cerevisiae*) was constructed by inserting an δ-amnolevulinic acid synthase coding gene *leu2* into the *hem1* locus of the BY4741 strain [*MATa*, *his3Δ1*, *met15Δ0*, *ura3Δ0*, *hem1::leu2*] as described previously [20,21]. The growth of *Δhem1* in response to haem or Ga (III) complex of the haem precursor protoporphyrin IX (GaPPIX; a toxic haem analogue) was measured in an established yeast spotting assay [18,22]. In brief, the coding sequence of *Hc-hrg-1* with a FLAG-tag was inserted into the pYES2-CT vector in the *Not* I site, with the coding sequence of *Ce-hrg-1* and *Ce-hrg-4* as well as the empty vector used as controls. Constructs were verified by sequencing and then individually transformed into the *Δhem1* yeast using the lithium acetate method [62]. Yeast colonies were

grown on 2% glucose synthetic complete (SC; -Ura/-LEU) plates supplemented with 250 μM 5-aminolevulinic acid (ALA; Sigma-Aldrich, St. Louis, USA) and streaked on to 2% raffinose SC (-Ura/-LEU) plates supplemented with 250 μM ALA to deplete glucose (SD; -Ura/-LEU) for 48–72 h. Prior to spotting, residual haem was removed by incubating yeast in 2% raffinose SC (-Ura/-LEU) medium for 18–24 h. Cells were then diluted in water ($OD_{600}$ = 0.2), and 5 μl 10-fold serial dilutions of each transformant were spotted on to 2% raffinose SC (-Ura/-LEU) plates supplemented with either 0.4% glucose and ALA (250 μM; positive control), haemin chloride (0.04, 2.5 and 10 μM), GaPPIX (0, 10 and 20 μM; Frontier Scientific) or water (negative control), and then incubated at 30°C for 2–5 days. Growth curves of $\Delta hem1$ was determined by measuring the $OD_{600}$ after culturing cells ($OD_{600}$ = 0.1) in SD (-Ura/-LEU) medium supplemented with 2.5 μM haem after 14, 16, 18, 20, 22 and 24 h. Single site-directed mutagenesis of *Hc-hrg-1* was performed on the pYES2-CT-*Hc*-HRG-1-FLAG vector by PCR amplification. Primers used for site mutagenesis are listed in S2 Table.

## Yeast two-hybrid assay

The Matchmaker Gold Yeast Two-Hybrid System (Takara Biomedical Technology, Beijing, China) was used to identify *Hc*-HRG-1-intercting proteins. The pGBKT7-*Hc*-HRG-1 recombinant yeast expression vector (Takara Biomedical Technology) was used as a bait to screen the *H. contortus* cDNA yeast library–constructed using the pGADT7 vector (Takara Biomedical Technology). The selection of colonies containing putative interacting proteins was carried out by plating on to synthetic 'dropout' medium using the manufacturer's protocol. The positive clones were picked for sequencing and library plasmid DNA isolation. The prey and bait vectors were co-transformed into Y2HGold (Takara Biomedical Technology) and then plated on to synthetic 'dropout' medium for further study.

## Glutathione-S-transferase (GST) pull-down

GST (negative control) or *Hc*-HRG-1-GST fusion protein expressed in HEK293T cells were incubated with GST-sefinose resins (Sangon Biotech, Shanghai, China) at 4°C for 2 h, with constant agitation. After 3–4 washes with PBS, the resins were incubated with the whole-cell lysates (WCLs) of cells transfected with pcDNA3.1-*Hc*-VHA-2-FLAG overnight at 4°C, under constant agitation. GST resins were collected by centrifugation at $5000 \times g$ for 1 min at 4°C, and washed thrice in PBS. GST resins and WCLs were processed for subsequent Western blot analysis using a specific antibody probe against either GST or the FLAG tag [63].

## Co-immunoprecipitation (Co-IP)

The *Hc-hrg-1* coding sequence with a HA tag and the *Hc-vha-2* sequence with a FLAG tag were cloned into the *Hind* III-*EcoR* I sites of the pcDNA3.1(+) vector (Takara Biomedical Technology). These plasmids were co-transfected into the HEK293T cells. At 24 h post-transfection, the cells were lysed in radioimmunoprecipitation (RIPA) lysis buffer (Fude Biological technology) containing $1 \times$ protease inhibitor cocktail (Bimake, Houston, USA) at 4°C for 30 min, with constant rocking. The cellular debris was removed by centrifugation at $12,000 \times g$ for 10 min at 4°C. The supernatant was incubated with 20 μl anti-FLAG magnetic beads (Bimake) overnight at 4°C, under constant agitation. Immunoprecipitated proteins bound to beads were concentrated using a magnetic separator, washed three times in PBST (NaCl 136.89 mM; KCl 2.67 mM; Na2HPO4 8.1 mM; KH2PO4 1.76 mM; 0.5% Tween-20) and eluted in 50μl sodium dodecyl sulfate (SDS) buffer (50 μl) containing (1.5% Tris, 9.4% glycine and 0.5% SDS). The immunoprecipitates and WCLs were analysed by Western blot probed with specific antibodies against HA or FLAG tag.

## Quantitative reverse transcription-PCR (qRT-PCR)

Total RNAs were extracted from the eggs, L1s, L2s, L3s, L4s (female and male) and adults (female and male) of *H. contortus*. The transcription of *Hc-hrg-1* in free-living and parasitic (blood-feeding) stages was determined by qRT-PCR employing SYBR Green real-time PCR master mix (Toyobo), performed on the LightCycle 480 system (Roche, Basel, Switzerland). Relative mRNA levels of *Hc-hrg-1* in different developmental stages were calculated using the $2^{-\Delta Ct}$ method. Each sample was tested in triplicate using the small subunit of the nuclear ribosomal RNA gene as an internal control [64]. Chemical knock-down of transcription was assessed in larvae (L1s and L3s) and adults (female and male) of *H. contortus* incubated with 0, 20 and 100 μM haemin chloride (Sigma-Aldrich, USA) in Roswell Park Memorial Institute (RPMI) 1640 medium containing 25 mM 4-(2-hydroxyethyl)-1-piperazineethanesulfonic acid, 1% FBS, 100 U/ml streptomycin, 10 U/ml penicillin at 28˚C for 24 h. Following incubation, worms were harvested, washed three times in PBS and snap-frozen in liquid nitrogen for subsequent RNA extraction. Transcriptional alterations of *Hc-hrg-1* in *H. contortus* in response to haemin chloride exposure was determined using the $2^{-\Delta\Delta Ct}$ method [65]. In qRT-PCR, at least three replicates of each sample were assessed.

## Double stranded RNA interference (RNAi)

In *C. elegans*, the coding sequence of *Ce-hrg-1* was cloned into the *Hind* III site of the L4440 RNAi vector using the NovoRec PCR Seamless Cloning Kit (Novoprotein). Recombinant plasmids were transformed into *E. coli* HT115 to yield *E. coli* expressing dsRNA targeting *Ce-hrg-1*. RNAi in *C. elegans* was performed by feeding using an established method [11]. In brief, equal number of synchronised L1s were placed on nematode growth media (NGM) agar plates containing 2 mM isopropyl β-D-1-thiogalactopyranoside (IPTG) and seeded with the transformed bacteria. *E. coli* HT115 bacteria transformed with parental L4440 vector is used as a negative control. Oligonucleotide primers used to produce double-stranded RNA are shown in S2 Table.

In *H. contortus*, a similar feeding method was employed for RNAi of *Hc-hrg-1* [66]. In brief, 2,000 eggs isolated from faeces were cultured in 3 ml of Earle's balanced salt soulution (EBSS) in a 25 cm$^2$ culture flask (Corning, NewYork, USA) for 7 days at 27˚C in 80% relative humidity. Bacteria expressing dsRNA targeting *Hc-hrg-1* were inoculated into the culture medium. *E. coli* expressing dsRNA targeting *cry1Ac* gene of *B. thuringiensis* (*Bt-cry1Ac*, GenBank accession number GU322939.1) were used as an "irrelevant" control [67]. *E. coli* containing the parental L4440-vector was used as a blank control. Worms were collected after three or seven days after treatment to measure the transcription of *Ce-hrg-1* by qRT-PCR. Phenotypes (development, motility and death) were recorded as described previously [11].

## Animal experimentation

Hu sheep (n = 12) were raised under a helminth-free condition and separated equally into three pens after birth. Eggs of *H. contortus* (n = 10,000) isolated from faeces were cultured in 3 ml culture medium (80% physiological saline, 19% EBSS, 1% yeast extract, 50 μg/mL ampicillin, 2 μg/mL amphotericin B, 5 μg/mL 5-fluorocytosine) in a 25 cm$^2$ flask (Corning, USA) seeded with *E. coli* HT115 (OD = 0.23–0.24) expressing dsRNA targeting *Hc-hrg-1*, for 7 days at 27˚C at 80% relative humidity. Larvae hatched from eggs (L1s) that had fed on bacteria for 7 days (L3s) were collected, as were larvae that had ingested *E. coli* expressing dsRNA targeting *Bt-cry1Ac*, or *E. coli* containing the parental L4440-vector were used as "irrelevant gene" and "blank" (i.e. negative) controls, respectively. Gene silencing of *hrg-1* in *H. contortus* was determined using qRT-PCR as described above. The three groups of sheep were infected with the

*hrg-1* gene-silenced L3s of *H. contortus*, irrelevant and blank controls, respectively. The number of eggs produced by adult worms and released into the faeces was conducted using the modified McMaster's faecal egg counting technique, after 21 days post infection [68]. Necropsy was performed after 40 days post infection to assess the infection burden in sheep.

## Statistical analysis

Data were analysed using GraghPad Prism 8 and shown as means ± standard deviation (SD) or means ± standard error of the mean (SEM). All statistical analyses were carried out using one-way ANOVA or Student's t test. A *P* value ≤ 0.05 was considered statistically significant.

## Supporting information

**S1 Table. A list of *hrg-1* orthologues identified in 41 parasitic nematodes of animals.**
(XLSX)

**S2 Table. Primer sets used in PCR for molecular cloning, heterologous expression, site-directed mutagenesis, yeast two-hybrid, pull-down, co-inmunoprecipitation, quantitative real-time PCR and/or RNA interference.**
(XLSX)

**S1 Fig. Haem transporter activity and associated key amino acid residues of *Hc*-HRG-1 in *Δhem1* yeast model.** (A) Heterologous protein expression of FLAG-tagged *Ce*-HRG-4 or *Hc*-HRG-1 in *Δhem1* yeast strain verified by Western blot analysis. (B) Heterologous protein expression of FLAG-tagged *Ce*-HRG-1/4 or *Hc*-HRG-1 in BY4741 yeast strain verified by Western blot analysis. (C) Heterologous protein expression of FLAG-tagged *Hc*-HRG-1 (H30A, H52A, H65A, H97A, H145A, Y29A, Y50A, Y55A, Y134A, Y136A, C2A, C24A, C70A, C73A and C89A) mutants in *Δhem1* yeast. (D) Spotting assay and growth curve of yeast transformed with empty vector, *Ce*-HRG-4, *Hc*-HRG-1 or *Hc*-HRG-1 mutants with or without supplementation with haem (2.5 μM or 10 μM) or 5-aminolevulinic acid (ALA–an intermediate in the haem biosynthesis). M: protein molecular weight marker.
(TIF)

**S2 Fig. Haem responsive element (HERE) prediction for *hrg-1* of *Haemonchus contortus* (*Hc-hrg-1*).** (A) Prediction of the HERE for *Hc-hrg-1* based on information on the haem responsive genes (e.g., *Ce-hrg-1* and *Ce-hrg-7*) from the free-living model organism *Caenorhabditis elegans*. (B) An image showing the ensheathed infective larvae of *H. contortus* that is transcriptionally non-responsive to haem supplementation. Sheath is indicted by red arrows.
(TIF)

**S3 Fig. Promoter activity test in *Caenorhabditis elegans*.** (A) Promoter of *hrg-1* (P$_{Ce-hrg-1}$) drives the expression of green fluorescent protein coding gene (*gfp*) in the anterior, medial and posterior intestine of *C. elegans*. (B) Promoter of lysosomal associated membrane protein 2 coding gene (P$_{Ce-lmp-2}$) drives gene expression in the posterior intestine of *C. elegans*. GFP: green fluorescent protein. DAPI: 4',6-diamidino-2-phenylindole.
(TIF)

## Acknowledgments

The Shared Management Platform for Large Instrument and The Experimental Teaching Center, College of Animal Science, Zhejiang University are greatly appreciated. We also acknowledge the technical assistance by Dr Yunqin Li from College of Animal Science, Zhejiang

University on laser confocal microscopy. Protein structure modelling was undertaken using the LIEF HPC-GPGPU Facility hosted at the University of Melbourne. We are grateful to Professor Caiyong Chen, College of Life Sciences and Innovation Center for Cell Signaling Network, Zhejiang University, China, for support during experiments and the drafting of this manuscript.

## Author Contributions

**Conceptualization:** Aifang Du, Guangxu Ma.

**Data curation:** Chaoqun Yao, Tao Wang, Robin B. Gasser, Guangxu Ma.

**Formal analysis:** Jingru Zhou, Fei Wu, Danni Tong.

**Funding acquisition:** Yi Yang, Aifang Du, Robin B. Gasser, Guangxu Ma.

**Investigation:** Yi Yang, Jingru Zhou, Fei Wu, Danni Tong, Shengjun Jiang, Yu Duan.

**Methodology:** Xueqiu Chen.

**Project administration:** Xueqiu Chen.

**Resources:** Yi Yang, Jingru Zhou, Xueqiu Chen.

**Supervision:** Guangxu Ma.

**Writing – review & editing:** Chaoqun Yao, Tao Wang, Aifang Du, Robin B. Gasser, Guangxu Ma.

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
