## [Decision Letter · Decision Letter 0]

22 Dec 2022

Dear Dr Ma,

Thank you very much for submitting your manuscript "Haem transporter HRG-1 is essential in the barber’s pole worm and an intervention target candidate" for consideration at PLOS Pathogens. As with all papers reviewed by the journal, your manuscript was reviewed by members of the editorial board and by several independent reviewers. In light of the reviews (below this email), we would like to invite the resubmission of a significantly-revised version that takes into account the reviewers' comments.

Thank you for submitting your manuscript to PlosPathogens and apologies for the long turnaround time in review. The manuscript was reviewed by three outside experts. All reviewers felt that the work is important and evaluates an important issue (haem uptake) in Haemonchus contortus. In addition to providing insights on the biology of this nematode, the finding of the essential nature of HRG-1 indicates a possible novel drug target for H. contortus and perhaps other nematodes as well.

That stated, the reviewers raise several issues that affect how well the study ultimately proves the conclusions. If you are able to address these issues in a timely manner, the revised manuscript will be re-considered for publication. If this is not possible, would consider submitting to another journal.

While all the comments from the reviewers should be addressed in a possible response, it would be critical to address the following concerns:

- Describe the in vitro binding data in the text.

- In figure 2B, please show (or mention in results) objective data of transformant growth when grown with ALA, and please show growth of normal (Wild-type) yeast alongside transformants with and without ALA supplementation.

- Show data that transformants with targeted mutations exhibit decreased growth. Currently the line stating this refers to figure 2D, which only shows a structural diagram.

- Improve labeling in figure 7.

- Add to methods the sequences used for cloning, qPCR, and siRNA experiments

- Add legends for the supplemental figures

- Expand the discussion to place the work in the context of prior studies.

- Given that the worms used to infect sheep were already dying at time of infection, the relevance of in vivo challenge of sheep is a bit unclear. At a minimum, would add language clearly stating the limitation of this experiment.

-Ed Mitre, academic editor, PlosPathogens

We cannot make any decision about publication until we have seen the revised manuscript and your response to the reviewers' comments. Your revised manuscript is also likely to be sent to reviewers for further evaluation.

Sincerely,

Edward Mitre

Academic Editor

PLOS Pathogens

P'ng Loke

Section Editor

PLOS Pathogens

Kasturi Haldar

Editor-in-Chief

PLOS Pathogens

orcid.org/0000-0001-5065-158X

Michael Malim

Editor-in-Chief

PLOS Pathogens

orcid.org/0000-0002-7699-2064

Thank you for submitting your manuscript to PlosPathogens and apologies for the long turnaround time in review. The manuscript was reviewed by three outside experts. All reviewers felt that the work is important and evaluates an important issue (haem uptake) in Haemonchus contortus. In addition to providing insights on the biology of this nematode, the finding of the essential nature of HRG-1 indicates a possible novel drug target for H. contortus and perhaps other nematodes as well.

That stated, the reviewers raise several issues that affect how well the study ultimately proves the conclusions. If you are able to address these issues in a timely manner, the revised manuscript will be re-considered for publication. If this is not possible, would consider submitting to another journal.

While all the comments from the reviewers should be addressed in a possible response, it would be critical to address the following concerns:

- Describe the in vitro binding data in the text.

- In figure 2B, please show (or mention in results) objective data of transformant growth when grown with ALA, and please show growth of normal (Wild-type) yeast alongside transformants with and without ALA supplementation.

- Show data that transformants with targeted mutations exhibit decreased growth. Currently the line stating this refers to figure 2D, which only shows a structural diagram.

- Improve labeling in figure 7.

- Add to methods the sequences used for cloning, qPCR, and siRNA experiments

- Add legends for the supplemental figures

- Expand the discussion to place the work in the context of prior studies.

- Given that the worms used to infect sheep were already dying at time of infection, the relevance of in vivo challenge of sheep is a bit unclear. At a minimum, would add language clearly stating the limitation of this experiment.

-Ed Mitre, academic editor, PlosPathogens

Reviewer's Responses to Questions

**Part I - Summary**

Reviewer #1: The manuscript by Yang and coworkers describes studies to characterize the function of a heme transporter in Haemonchus contortus. The work is thorough and important. The manuscript is well written.

Reviewer #2: The article by Yang et al describes the role of the heme transporter HRG-1 is Haemonchus contortus using molecular approaches in yest, C. elegans and H.contortus itself. Given nematodes can not de novo synthetise heme but need to acquire it from their environment, HRG-1 is a potential target of interest for sustainable control of Hc. The authors showns that nematode parasite have only one HRG-1, that it is associated with endosome/lysosome and that RNAi decrease parasite survival. Alltogether, there is substantial evidence that moving forward HRG-1 as a drug or vaccine target could be of interest.

Reviewer #3: This is important work that focuses on a still understudied parasite despite its veterinary importance. I liked aspects of the study but I found major issues as well.

STRENGTHS: Much of the study appears novel, and represents an ambitious attempt to study haem uptake in Hc in some detail. In particular the animal experiments are compelling.

WEAKNESSES: I have concerns around execution and interpretation of some of the data, as described in more detail below.

**Part II – Major Issues: Key Experiments Required for Acceptance**

Reviewer #1: (No Response)

Reviewer #2: Line 188 : This point is in apparent contradiction with figure 6A, where the expression is found to be higher in males in the adult stage. Interestingly the data are not provided for adult females. Rather than having this comment based on one representative image, it would be interesting to have an actual quantification on HRG1 expression in gonads of male and female to ensure that this point is correct. Of note, in the male section, the intestine expression of HRG1 also seems much weaker than in the female, the difference might thus more due to an suboptimal choice of representative image.

- For figure 7: It would have made comparison between parasites much easier to have similar staining presented (section versus whole worm). I am in particular not sure whether the authors can claim a "basal laminae" staining based on those images. Why would it be "apical membrane" for Ce and "basal laminae" for Hc for intestine? The intestinal staining in Hc certainly looks in the intstinal epithelium itself, as it is quite diffuse around the nuclei of the intestinal epithelium. Due to high autofluorescence of many nematode tissues, the authors should present images of non GFP parasite to confirm specificity of HRG-1 localisation.

Reviewer #3: I have the following concerns:

LINE 109: Refers to in vitro experiments as already having been completed, without any previous description of these or presentation of data. Associated Figure 1D shows in vitro binding data but these data are not described in the results text.

LINE 113: "Grew normally" - is a dubious claim. The Hc-HRG-1 transformant doesn't look very different to my eyes from the empty vector. I would ask for these experiments to include a wild type +ve control to assess whether growth is "normal". Evaluation of these results as significant is currently subjective; yeast growth images need to be quantified by densitometry or similar, with statistical analysis of data.

LINE 123: Claims that transformants carrying targeted mutations exhibit decreased growth, referring to Figure 2D. Figure 2D is merely a structural diagram, so the data supporting this claim is not shown.

LINE 130: Co-localisation of HRG/ZnMP refers to Fig 3B subpanel. Red fluorescence in these panels is not clear, making it difficult to interpret the co-localisation claim.

LINE 167: States that "In this C. elegans model, Hc-hrg-1 RNA interference (RNAi)-mediated (heterologus) gene knockdown resulted in a significant (P < 0.01) decrease in the Ce-hrg-1 mRNA level, being comparable to homologous knockdown of Ce-hrg-1 (P < 0.01) (Fig 6C)", which is a confusing statement. Does this imply that the Ce were expressing Hc-HRG? Or that Ce were treated with Hc-HRG dsRNA? This section needs to be written more clearly. AS a non-specialist it is not clear to me why HRG knockdown INCREASES ZNP fluorescence? Shouldn't this impede ZNP uptake leading to REDUCED fluorescence? Again, this needs to be more clearly explained.

LINE 181, and Figure 7B: How do we know Hc localisation is gonadal? Clearer labelling and/or additional evidence is required to demonstrate that the observed fluorescence is associated with reproductive tissue. Also the diagram in Figure 7D could be useful in this regard but currently doesn't illustrate reproductive expression at all.

MOLECULAR METHODS: No primer sequences have been defined, making replication of cloning, qPCR and RNAi experiments impossible.

Fig 5 legend: Meaning of "HA" in Hc-HRG-1-HA is not defined.

LINE 199: Refers to RNAi phenotypes including "dead" and "sick" worms. These are subjective terms, and should be quantified and explained more clearly.

**Part III – Minor Issues: Editorial and Data Presentation Modifications**

Reviewer #1: • More explanation of methods used in each part of the results would be useful. Not detailed methods, but a general overview of what was done. For example, line 109, it is stated that Hc-hrg-1 binds heme in vitro; explain how this was determined?

• Line 113: the Hc-HRG-1 transformed cells maybe grew a little better than dhem1 cells alone, but not nearly that of Ce-HRG-4 transformed cells. Why was Ce-HRG-4 and not HRG-1 used? The GaPPIX treatment was more convincing.

• Line 121, mutagenesis: Were the mutant proteins expressed and nonfunctional or simply not expressed or unstable?

• The yeast 2-hybrid screen identified only the V-ATPase and no other binding partners?

• Comment on the L3 having highest Hc-hrg-1 mRNA, but exogenous heme was not accessible due to ensheathment. What is the function of Hc-hrg-1 in this stage?

• Line 166: The way I read this Hc-hrg-1 RNA was used to silence Ce-hrg-1 in Ce. If there is similarity between Hc and Ce this is expected. What about Ce-hrg-1 paralogues in Ce? Would these also be silenced by Hc-hrg-1 treatment? Is this the same result seen in Ce silencing with Ce sequences?

• Line 174: Silencing both hrg-1 and vha-1 lead to an increase in hrg-1 mRNA?

• Line 187: It is indicated that heme in excess is toxic. How does Hc deal with excess heme?

• Line 203: Please explain how adding exogenous heme rescues worms that have had the protein that acquires exogenous heme silenced.

• Ce has 4 paralogues of hrg-1. These should be discussed in more detail.

• Fig. 8. E and F are after 3 d treatment and G and H after 7 ay treatment? Sick + dead after 7 d treatment is about 60%. These were used to infect sheep and patent infection was not observed. Is this surprising since most of the worms were already dead or dying? This questions the usefulness of the in vivo studies.

• Discussion of the results of the study was only 2 paragraphs. This should be expanded. Place the results in context of previous studies. For example, how do the results of this study compare to the study of heme uptake/transport in Ce and Brugia?

• Are there legends for the supplemental figures/table? Was is the meaning of Target %id and Query %id in SI table (2)?

Reviewer #2: - The english is not always clear. In particular lengthy sentences, make understanding very difficult. Please see attached document to see details of which sentence need changes.

- Figure 1A : What species have actually been investigated ? Were there any that did not have HRG-1 homologues ? It would of interest if Figure S1 could answer those questions

-line 99-101 : the authors describe that HRG-1 homology to human orthologues is limited, but pointed out that "particular amino acid (aa) residues in sequences linked to haem transport were invariable among the nematode and mammalian species"  could the authors please speculates on whether this would be a problem for targeting HRG-1 for example for vaccine design.

- Line 153: I do not understand the cause/effect link the authors are referring to. It would be help to rephrase/explain more clearly what the authors mean.

-Line 166: please define what the "Haem aquisition model" is.

-Figure 6a : The authors speculate that the high transcription of HRG-1 after heme exposure in L3 is linked to unresponsiveness to heme as L3 is a closed stage. However, I do not understand why it is so high in L3 and not in other free living stages. To me it looks like it is massively produced at the infective stage as it is highly required early on for infection ? It is really easy in vitro to activate iL3 with CO2 and 37C, and it would be interesting to see if the parasitic L3 stage is more similar to the infectious or to L4 to understand the role of HRG-1, as the authors later claim in the paper that "knock down" of HRG-1 prevent parasite establishment.

- Figure 8: I respectfully disagree with the authors interpretation of the data presented. Please justify/discuss this point and perform appropriate statistical test (see comments in the pdf line 194 and 203)

- Line 218: Nothing in the experiment presented can point out that the lower adult worm/egg outpout burden at endpoint is linked to a decrease establishment of parasite. It could either be lower establishment or lower survival/reproduction. Earlier time point is required for making this claim. Please rephrase accordingly or add the new experiment.

Reviewer #3: N/A

PLOS authors have the option to publish the peer review history of their article (what does this mean?). If published, this will include your full peer review and any attached files.

Reviewer #1: No

Reviewer #2: No

Reviewer #3: No
---

## [Editor Report · Decision Letter 1]

18 Jan 2023

Dear Dr Ma,

We are pleased to inform you that your manuscript 'Haem transporter HRG-1 is essential in the barber’s pole worm and an intervention target candidate' has been provisionally accepted for publication in PLOS Pathogens.

Best regards,

Edward Mitre

Academic Editor

PLOS Pathogens

P'ng Loke

Section Editor

PLOS Pathogens

Kasturi Haldar

Editor-in-Chief

PLOS Pathogens

orcid.org/0000-0001-5065-158X

Michael Malim

Editor-in-Chief

PLOS Pathogens

orcid.org/0000-0002-7699-2064
---

## [Editor Report · Acceptance letter]

26 Jan 2023

Dear Dr Ma,

We are delighted to inform you that your manuscript, "Haem transporter HRG-1 is essential in the barber’s pole worm and an intervention target candidate," has been formally accepted for publication in PLOS Pathogens.

Best regards,

Kasturi Haldar

Editor-in-Chief

PLOS Pathogens

orcid.org/0000-0001-5065-158X

Michael Malim

Editor-in-Chief

PLOS Pathogens

orcid.org/0000-0002-7699-2064